# Unconventional CN vacancies suppress iron-leaching in Prussian blue analogue pre-catalyst for boosted oxygen evolution catalysis

Zi-You Yu[1,5], Yu Duan[1,5], Jian-Dang Liu[2,5], Yu Chen[1], Xiao-Kang Liu[3], Wei Liu [3], Tao Ma[1], Yi Li[1], Xu-Sheng Zheng[3], Tao Yao [3], Min-Rui Gao [1], Jun-Fa Zhu [3], Bang-Jiao Ye[2] & Shu-Hong Yu [1,4]

The incorporation of defects, such as vacancies, into functional materials could substantially tailor their intrinsic properties. Progress in vacancy chemistry has enabled advances in many technological applications, but creating new type of vacancies in existing material system remains a big challenge. We show here that ionized nitrogen plasma can break bonds of iron-carbon-nitrogen-nickel units in nickel-iron Prussian blue analogues, forming unconventional carbon-nitrogen vacancies. We study oxygen evolution reaction on the carbon-nitrogen vacancy-mediated Prussian blue analogues, which exhibit a low overpotential of 283 millivolts at 10 milliamperes per square centimeter in alkali, far exceeding that of original Prussian blue analogues and previously reported oxygen evolution catalysts with vacancies. We ascribe this enhancement to the in-situ generated nickel-iron oxy(hydroxide) active layer during oxygen evolution reaction, where the Fe leaching was significantly suppressed by the unconventional carbon-nitrogen vacancies. This work opens up opportunities for producing vacancy defects in nanomaterials for broad applications.

[1] Division of Nanomaterials & Chemistry, Hefei National Laboratory for Physical Sciences at the Microscale, CAS Center for Excellence in Nanoscience, Hefei Science Center of CAS, Collaborative Innovation Center of Suzhou Nano Science and Technology, Department of Chemistry, University of Science and Technology of China, Hefei 230026, China. [2] State Key Laboratory of Particle Detection and Electronics, University of Science and Technology of China, Hefei 230026, China. [3] National Synchrotron Radiation Laboratory, University of Science and Technology of China, Hefei 230029, China. [4] Dalian National Laboratory for Clean Energy, Dalian 116023, China. [5] These authors contributed equally: Zi-You Yu, Yu Duan, Jian-Dang Liu. Correspondence and requests for materials should be addressed to M.-R.G. (email: mgao@ustc.edu.cn) or to S.-H.Y. (email: shyu@ustc.edu.cn)

Prussian blue analogues (PBAs) are a class of perovskite-type materials described by the general formula $A_xM[Fe(CN)_6]_y \cdot mH_2O (0 \leq x \leq 2, y<1)$, where A is alkali metal and M is transition metal[1–3]. In such materials, nitrogen-coordinated M cations and carbon-coordinated Fe sites are bridged by cyanide (CN) groups, forming open frameworks. Research on PBAs reveals that their intriguing structures enable diverse applications, such as molecule-based magnets[4,5], sensors[6,7], hydrogen storage[8,9] and metal-ion batteries[1,10–12]. In recent years, PBAs have also emerged as catalytic materials for oxygen evolution reaction (OER)[13–15]—a key enabling process for rechargeable metal–air batteries and photo/electrochemical water splitting[16,17]. For examples, Galan-Mascaros et al.[14] described decent OER activity on Co–Fe PBA catalyst, which was later found to be stable in acid through suitable chemical etching treatment[14]. Zhang et al. used operando X-ray spectroscopy on Ni–Fe PBA to identify the catalytically active sites, suggesting that amorphous Ni hydroxide after OER contributes to the reactivity[15]. Nevertheless, the role of Fe species in the PBA catalysts for OER is unclear. Moreover, engineering the structure of these CN-bridged bimetallic open frameworks may lead to further enhancements in OER catalysis, which, however, has rarely been researched.

Manipulating defect chemistry can tune the properties and functionalities of materials, such as band structure, conductivity, magnetism, and catalysis[18–20]. The most common defect observed is vacancy, and typically, vacancy engineering can lead to substantial structural perturbations in catalysts, with the capability to tailor surface electronegativity, charge concentration, and redistribution[18,19]. Once overcoming their formation energy, both anion and/or cation vacancies can be formed, giving diverse vacancy defects like oxygen[21,22], sulfur[20,23], iodine[24], nickel[25], iron[26], and sometimes dual vacancies[27,28]. PBAs are the well-known materials bearing $Fe(CN)_6$ vacancies that permit the alteration of electron transfer phenomena[1,2]. In principle, a new type of vacancy created in PBAs would trigger the change of local electronic environment, which subsequently leads to regions of enhanced energetics for catalysis. But no previous attempts were seen to produce unconventional vacancies in PBA materials, mainly owing to the lack of effective synthetic pathways.

Here, we report the generation of unusual CN vacancies (denoted as $V_{CN}$) in the Ni–Fe PBA (i.e., $K_2NiFe(CN)_6$) through a nitrogen plasma bombardment. Despite the strong affinity of metals and CN ligands, our comprehensive characterizations confirm that the ionized $N_2$ can overcome the energy of $V_{CN}$ formation, yielding unprecedented $V_{CN}$. The obtained Ni–Fe PBA catalyst with $V_{CN}$ enables highly efficient OER catalysis, far exceeding that of original sample and also previously reported vacancy-mediated OER catalysts. Besides offering tailored local electronic features, our another key finding is that the presence of $V_{CN}$ can suppress the release of Fe species, which therefore favors the growth of Ni–Fe oxy(hydroxide) active layer via a dynamic self-reconstruction of the PBA pre-catalyst during OER. Our work widens the family of vacancy defects, which provides access to enhanced functionalities of PBA materials for various applications.

## Results

### Synthesis and characterization of $V_{CN}$-mediated Ni–Fe PBA.
We started with the synthesis of $NiMoO_4$ nanorods acted as precursors using a hydrothermal method described by us previously[29,30] (see the Methods section; Supplementary Fig. 1). The $NiMoO_4$ was then dispersed in a $K_4Fe(CN)_6 \cdot 3H_2O$ aqueous solution under ultrasonication for 2 h, which templates the formation of one-dimensional $K_2NiFe(CN)_6$ Prussian blue analogues (PBAs;

Supplementary Fig. 1). Consequently, we used $N_2$ plasma bombardment to trigger the $V_{CN}$ formation, as shown schematically in Fig. 1. This process provides energetic nitrogen radicals that enable the cleavage of Fe–C and Ni–N bonds in Fe–CN–Ni units, whereas cleaving the CN triple bond needs to overcome a higher activation barrier; hence the generated $CN^-$ diffuses away from the PBA lattices, forming $V_{CN}$ (Fig. 1; Supplementary Fig. 2). Figure 2a shows the transmission electron microscopy (TEM) image of the PBA bombarded with $N_2$ plasma for 60 min (which we abbreviate PBA-60), revealing the rod-like structure with jagged edges. Close-up inspection of these polycrystalline nanorods (inset in Fig. 2a) shows that they actually consist of many interconnected, welded cubes (Supplementary Fig. 3). X-ray diffraction (XRD) analysis uncovers the cubic $K_2NiFe(CN)_6$ phase of the pristine PBA (JCPDS 200915; Supplementary Fig. 4). We also bombarded the PBAs with $N_2$ plasma for different times, and do not see noticeable signs of phase, component, or morphological changes compared with the pristine materials (Supplementary Figs 5, 6). These results together indicate that $N_2$ plasma bombardment would not cause structural damage of PBAs, but generates CN vacancies. The Brunauer–Emmett–Teller (BET) surface area of the PBA-60 was determined to be $68.3 m^2 g^{-1}$, larger than the value of pristine PBAs (i.e., PBA-0) owing to the more accessible vacancy sites (Supplementary Fig. 7).

We applied multiple characterization techniques to verify the formation of $V_{CN}$ in PBAs after $N_2$ plasma bombardment. Figure 2b gives a high-resolution TEM (HRTEM) image of PBA-60, which displays crystalline domains with discontinuous atomic arrangement, as further demonstrated by the peak valleys of the atomic intensity profile (inset in Fig. 2b). The local lattice discontinuity suggests the presence of $V_{CN}$ that created by ionized $N_2$ radicals. Scanning TEM (STEM) elemental mapping shows no clear C and N fades throughout the PBA-60 nanorods (Fig. 2c), which reveals that the concentration of $V_{CN}$ does not notably alter the bulk components. We further confirmed the $V_{CN}$ through positron annihilation spectroscopy (PAS), which can offer straightforward structure information of vacancy defects[31]. In Fig. 2d, the positron lifetime spectra of both PBA-0 and PBA-60 yield three lifetime components, where the two longer components ($\tau_2$ and $\tau_3$) come from the large voids and the interface in the materials (Supplementary Table 1). The measured shorter component ($\tau_1$) of 267 ps for PBA-60 matches well with the calculated positron lifetime of 270 ps for $V_{CN}$, implying that positron annihilation traps at such vacancies (Supplementary Tables 1, 2). Moreover, our calculation reveals that positrons mainly concentrate at the center of $V_{CN}$ in PBA-60 (Fig. 2e, f).

Considering that vacancy defects are able to tune the photoluminescence (PL) and cathodoluminescence (CL) properties through the mediated band-to-band transition[32], we conducted PL and CL measurements on PBA-0 and PBA-60 for comparison. We found clear PL and CL enhancements of PBA-60 versus PBA-0, probably owing to the high quantum efficiency of excitons localized at the $V_{CN}$ sites (Supplementary Figs 8, 9). In addition, we bubbled the ninhydrin and $Na_2CO_3$ aqueous solution with tail gas produced during the synthesis process, and observed the fading of original yellow solution, adding support to the formation of CN anions[33] (Supplementary Fig. 10). Together, all the above results provide solid evidences that CN vacancies are formed in PBA materials.

### OER performance.
To explore the influence of new CN vacancies on the OER catalytic behavior, we evaluated the OER activities of PBA-0 and $V_{CN}$-mediated PBAs in $O_2$-saturated 1 M KOH at ambient environment. The PBAs that bombarded by $N_2$ plasma for 10, 30, 120 min are abbreviated hereafter as PBA-10, PBA-30,

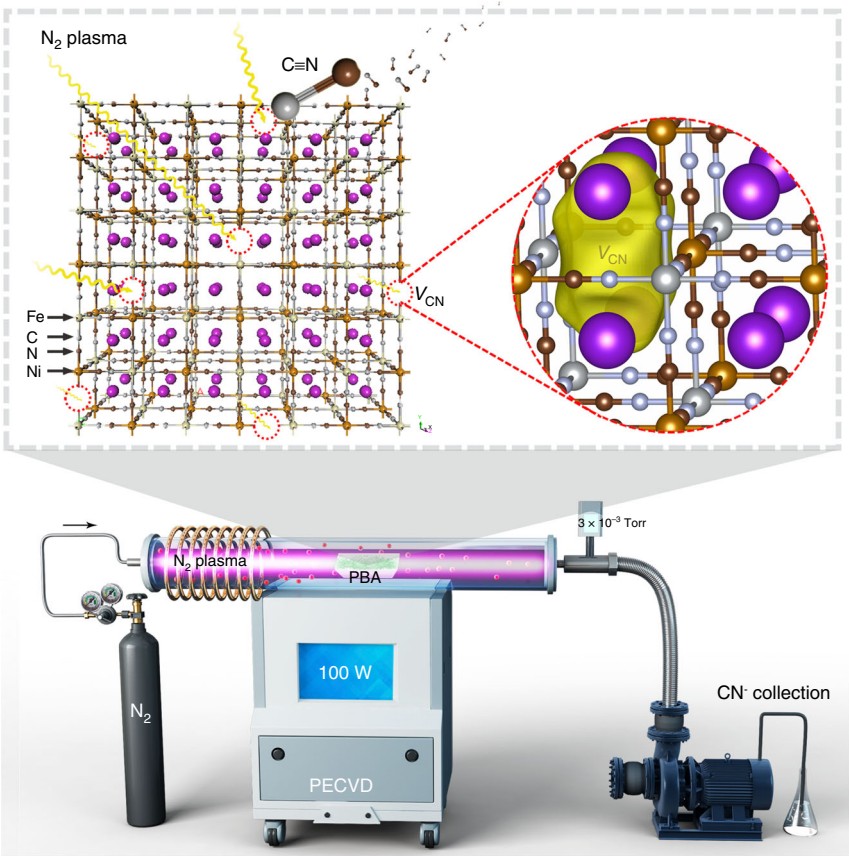

**Fig. 1** Schematic illustration of the preparation of $V_{CN}$-mediated Ni–Fe PBA. Top: $V_{CN}$ forms in Ni–Fe PBA material through $N_2$ plasma bombardment. Bottom: a full diagram of the $N_2$ plasma apparatus

and PBA-120. We recorded the OER polarization curves for comparison until the activity becomes stable (Supplementary Fig. 11). Figure 3a shows that all the PBA pre-catalysts exhibit a distinct peak at ~1.38 V originating from the $Ni^{2+}/Ni^{3+}$ redox process[34,35]. We found that PBA-0 requires an overpotential of 440 mV at 10 mA cm$^{-2}$ (normalized on the basis of geometrical surface area), which decreased to 368 mV for PBA-10, 321 mV for PBA-30, 283 mV for PBA-60, and 310 mV for PBA-120 (Fig. 3a). Thus, the formation of $V_{CN}$ in PBA plays a key role for improving the OER energetics. Note that the small overpotential of 283 mV at 10 mA cm$^{-2}$ for PBA-60 is 32 mV and 50 mV lower than that of the state-of-the-art NiFe-layered double-hydroxide (NiFe–LDH)[36] and 20 wt% Ir/C catalysts (Fig. 3a; Supplementary Figs 12, 13), respectively. Tafel plots (Fig. 3b) indicate that PBA-60 has a very small Tafel slope of 54 mV dec$^{-1}$, lower than 108 and 69 mV dec$^{-1}$ for PBA-0 and NiFe–LDH catalysts, respectively. Meanwhile, at the overpotential of 300 mV, PBA-60 has a 31-fold improvement in current density versus PBA-0, although the increase in BET surface area by a factor of mere ~1.2 (Supplementary Fig. 7). The predominant OER properties of PBA-60 over other studied catalysts are also clearly seen at overpotentials of 320 mV and 340 mV (Fig. 3c). These results again suggest that $V_{CN}$ should be the main reason that results in the remarkable OER energetics of PBA-60.

To probe the charge transfer resistance ($R_{ct}$) of studied catalysts, we conducted electrochemical impedance spectroscopy (EIS) at 400 mV overpotential, which offers the information regarding the number of electrons transferred from the catalyst surface to the reactant. Figure 3d reveals that the $R_{ct}$ of PBA-60 is 11 Ohms, substantially smaller than that of ~74 Ohms for PBA-0 and ~32 Ohms for the NiFe–LDH catalyst (Supplementary

Fig. 13b). This result matches well with our double-layer capacitance ($C_{dl}$) measurements that give the highest $C_{dl}$ value for PBA-60 catalyst (Supplementary Fig. 14). We then studied the influence of temperature on the OER performance of different catalysts at an overpotential of 300 mV to assess their kinetic barriers[16]. We observed a linear relationship between 293 K and 323 K that follows the Arrhenius behavior, from which the activation energies of 67 kJ mol$^{-1}$, 36 kJ mol$^{-1}$, and 44 kJ mol$^{-1}$ were extracted for PBA-0, PBA-60, and NiFe–LDH catalysts, respectively (Fig. 3e; Supplementary Fig. 15). The lowest apparent barrier value measured for PBA-60 provides evidence that OER kinetics on PBA is highly promoted by CN vacancies. In addition, the effect of plasma treatment atmosphere and power on OER activity showed that PBA-60 obtained by $N_2$ plasma bombardment at a power of 100 W has the optimal OER activity (Supplementary Figs 16–19).

Compared with previously reported catalysts with diverse vacancy defects, we underscore that our PBA-60 shows the best OER performances in terms of the overpotential at 10 mA cm$^{-2}$ and the onset than were shown before (Fig. 3f; Supplementary Table 3). We believe that the new $V_{CN}$ engineering of other materials can guarantee predictable better-performing OER catalysts.

**Chemical and structure alterations**. We now study the component evolution of PBA with time during the $N_2$ plasma bombardment. We applied the inductively coupled plasma atomic emission spectroscopy (ICP-AES; measuring K, Fe, and Ni) and elemental analysis (measuring C and N) to probe the bulk element variation, whereas the surface component evolution was determined by the X-ray photoelectron spectroscopy (XPS). We

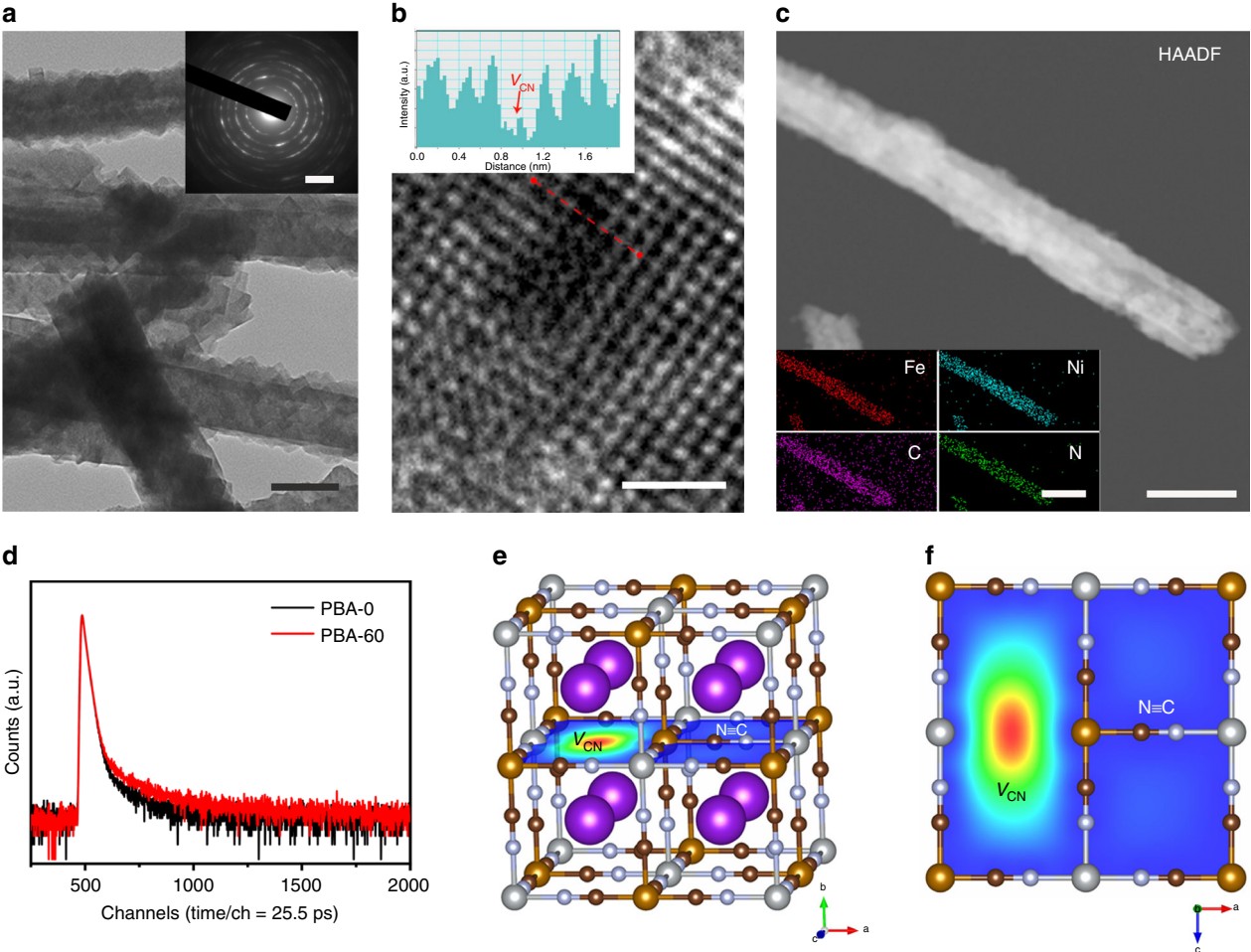

**Fig. 2** Characterization of $V_{CN}$-mediated Ni–Fe PBA. **a** TEM image of PBA-60. Scale bar, 100 nm. Inset shows the corresponding selected-area electron diffraction pattern. Scale bar, 2 nm$^{-1}$. **b** HRTEM image of PBA-60. Scale bar, 1 nm. Inset shows the atomic intensity profile along the dotted red line in **b**. **c** HAADF image of a typical PBA-60 nanorod. Scale bar, 50 nm. Inset shows the corresponding STEM elemental mappings. Scale bar, 100 nm. **d** PAS analysis of PBA-0 and PBA-60. **e, f** Schematic representation of the trapped positrons by $V_{CN}$-mediated PBA-60

found that both bulk and surface atomic ratios of K/Ni, Fe/Ni, and C/N remain unchanged even bombarding for 60 min, but the atomic ratios of N/Ni and C/Fe decreased with time (Supplementary Fig. 20, Supplementary Tables 4, 5). This unambiguously suggests the formation of sole $V_{CN}$ in PBA without any other vacancy defects. However, when we bombarded the PBA for 120 min, we observed a decreased Fe/Ni atomic ratio, indicative of some Fe vacancies formed in the structure (Supplementary Fig. 20). Because no Ni loss during $N_2$ plasma process, we are able to calculate the $V_{CN}$ content of different samples based on the measured N/Ni atomic ratios. The bulk $V_{CN}$ contents of ~3.2, 4.5, 7.5, and 11.4% are determined for PBA-10, PBA-30, PBA-60, and PBA-120, slightly lower than that of surface $V_{CN}$ contents (Fig. 4a). Moreover, our Ni and Fe K-edge extended X-ray absorption fine structure (EXAFS) fittings reveal that, after bombarding with $N_2$ plasma for 60 min, the first-shell Ni–N coordination number decreases from 5.9 to 5.5 and the Fe–C coordination number decreases from 6.1 to 5.7 (Fig. 4b; Supplementary Figs 21–24, Supplementary Table 6). Such unsaturated local atomic environment that comes from the formation of $V_{CN}$ could promote the energetics of the OER[18].

To further understand the source of high OER activity, we monitored the oxidation states and electronic structure of PBA catalysts using Raman spectroscopy, X-ray absorption spectroscopy (XAS), and XPS. Our Raman spectroscopy measurement on PBA-0 (Fig. 4c) exhibits two prominent peaks at 2098 cm$^{-1}$ and 2134 cm$^{-1}$, corresponding to the vibrations of CN groups in the mixture of Fe$^{II}$–CN–Ni$^{II}$ and Fe$^{II}$–CN–Ni$^{III}$, whereas the peak at 2225 cm$^{-1}$ comes from the CN vibration in Fe$^{III}$–CN–Ni$^{II}$, in agreement with previous reports[10,11]. We found that the peak at 2225 cm$^{-1}$ disappears for PBA-60 sample, meaning the changed oxidation state of Fe$^{III}$ species. Because XAS technique is sensitive to the local structure and chemical environment, we performed Ni L-edge XAS measurements on different PBA samples with NiO and LiNiO$_2$ as references (Supplementary Fig. 25). As Fig. 4d shows, the Ni L$_3$-edge shoulder peak shifts from 855.1 eV (PBA-0) to 855.6 eV (PBA-60), indicating that surface Ni$^{2+}$ ions were partially oxidized to Ni$^{3+}$ after $N_2$ plasma bombardment[37,38] (Supplementary Fig. 25). The Fe L$_3$-edge XAS spectra in Fig. 4e and Supplementary Fig. 26 reveal two peaks that originated from electron transitions to unoccupied e$_g$ orbitals and the additional transitions to anti-bonding $\pi^*$ states[39] (inset in Fig. 4e). The increased intensity of the left peak demonstrates lowered oxidation state of Fe with the formation of $V_{CN}$. The above oxidation-state analyses match well with our XPS studies presented in Supplementary Fig. 27. Further, electron spin resonance (ESR) measurements show that an ESR signal appears

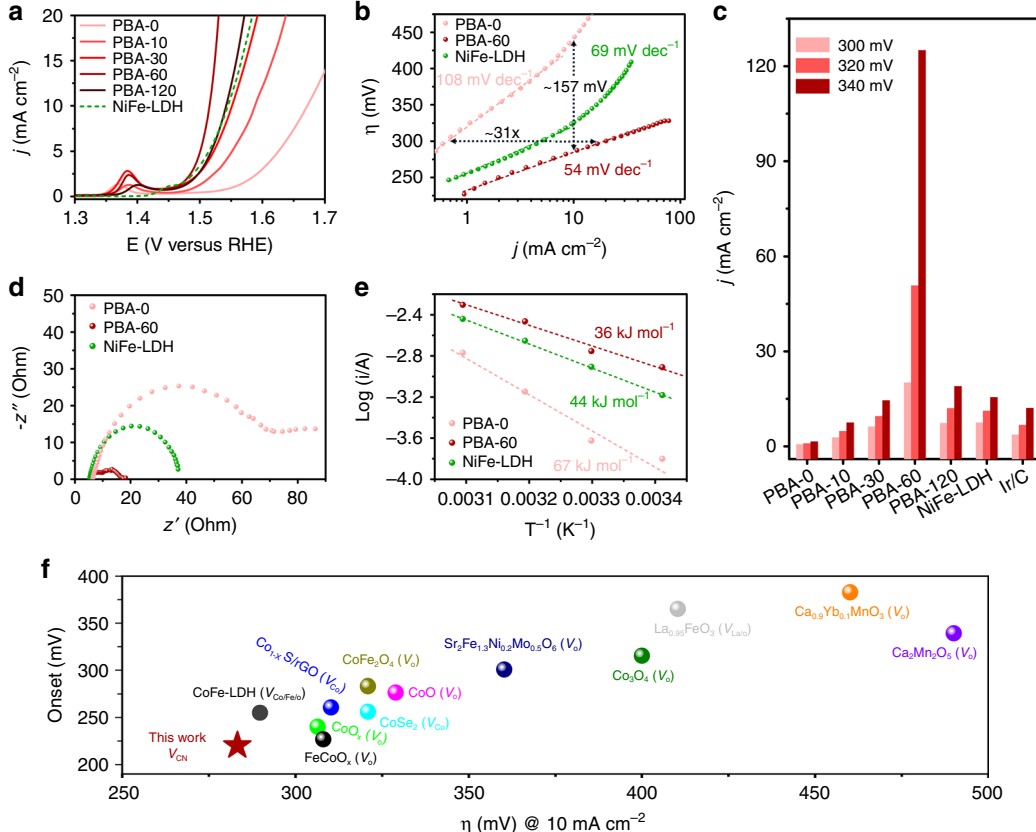

**Fig. 3** OER evaluation. **a** OER polarization curves of different studied catalysts. Catalyst loading: ~0.255 mg cm$^{-2}$. Sweep rate: 5 mV s$^{-1}$. **b** Tafel plots of different catalysts derived from polarization curves shown in Supplementary Fig. 13. Tafel plot of NiFe–LDH derived from the polarization curve of the cathodic sweep to avoid the interference of the redox peak. **c** Comparison of the current densities generated on different catalysts at overpotentials of 300 mV, 320 mV, and 340 mV, respectively. The data were extracted from the polarization curves shown in Supplementary Fig. 13. **d** EIS Nyquist plots of different catalysts at 400 mV overpotential without iR-correction. **e** Arrhenius plot of the OER kinetic current on different catalysts at 300 mV overpotential without iR-correction. **f** Comparison of onset potential (defined as the potential at 1 mA cm$^{-2}$) and overpotential at 10 mA cm$^{-2}$ for various OER catalysts with vacancy defects. Values were plotted from references where they are reported as such (Supplementary Table 3). All OER measurements were performed in O$_2$-saturated 1 M KOH electrolyte, and the reported data were iR-corrected unless otherwise stated

at $g$-factor = 2.03 just after 10 min of bombardment, which becomes dominant when bombarding for 60 min (Fig. 4f). This symmetrical ESR signal originates from the unpaired electrons of Ni$^{3+}$ (t$_{2g}^6$e$_g^1$) species[25,40], again indicating the increase in Ni oxidation state after incorporating the CN vacancies.

We now come to an understanding that the new $V_{CN}$ sites not only cause local unsaturated coordination environment but also modulate the oxidation states of Ni and Fe of PBAs. We next use the accumulated knowledge on Ni–Fe oxide OER catalysts to propose potential OER enhancement mechanism of our Ni–Fe PBAs, considering the similarity in their real active sites. Recent studies have demonstrated that Fe sites in Ni–Fe oxides are active, while Ni species act as electrically conductive and chemically stable host for the Fe sites[41,42]. In our fresh PBA-0 catalyst (i.e., K$_2$NiFe(CN)$_6$), both Ni and Fe are in the oxidation state of 2 + (Fig. 4d; Supplementary Fig. 28)[11]. The partial metastable Fe$^{2+}$ is readily oxidized to Fe$^{3+}$ in the lab environment[11,43], which returns to its original state after the $V_{CN}$ formed (Fig. 4c, e). However, the oxidation state of Ni in PBAs was partially boosted to 3 + through $V_{CN}$ mediation (Fig. 4d, f), which suggests the electron transfer from Ni to the adjacent Fe sites. During OER, the $V_{CN}$-mediated Ni–Fe PBA catalyst with modified metal oxidation state thus energetically transfers to NiFeOOH active layer via surface self-reconstruction (discuss later), giving rise to the marked OER activity.

Our experiments reveal that the OER activity of $V_{CN}$-mediated Ni–Fe PBA catalyst improves monotonously until the bombarding time reaches 60 min, where it gains a bulk $V_{CN}$ content of ~7.5% (Figs 3a, 4a). Further extending the bombarding time, however, causes the loss of Fe in the structure (Supplementary Fig. 20 and Tables 4, 5), leading to inferior OER properties. We further prepared Co–Fe PBA nanorods using the same methodology and created $V_{CN}$ through N$_2$ plasma bombardment (Supplementary Fig. 29). We also observed the $V_{CN}$-mediated OER improvement on Co–Fe PBAs, but its activity is largely lower than that of Ni–Fe PBAs (Supplementary Fig. 30), which demonstrates that Ni–Fe-based OER catalysts are notable, agreeing with previous reports[44].

**Fe leaching suppression by the $V_{CN}$.** Figure 5 illustrates another critical observation that we want to demonstrate in this work: that is, the Fe loss in Ni–Fe PBA catalysts is substantially suppressed in the presence of CN vacancies. Researchers have previously seen that [Fe(CN)$_6$]$^{4-}$ groups in PBAs readily diffuse outward to the electrolytes during OER, causing Fe loss in the structure[14,15]. Our operational stability assessments exhibit that PBA-60 catalyst performs very robustly, even at a high current density of 1 A cm$^{-2}$, whereas the PBA-0 decays gradually with time (Fig. 5a; Supplementary Fig. 31). Energy-dispersive X-ray

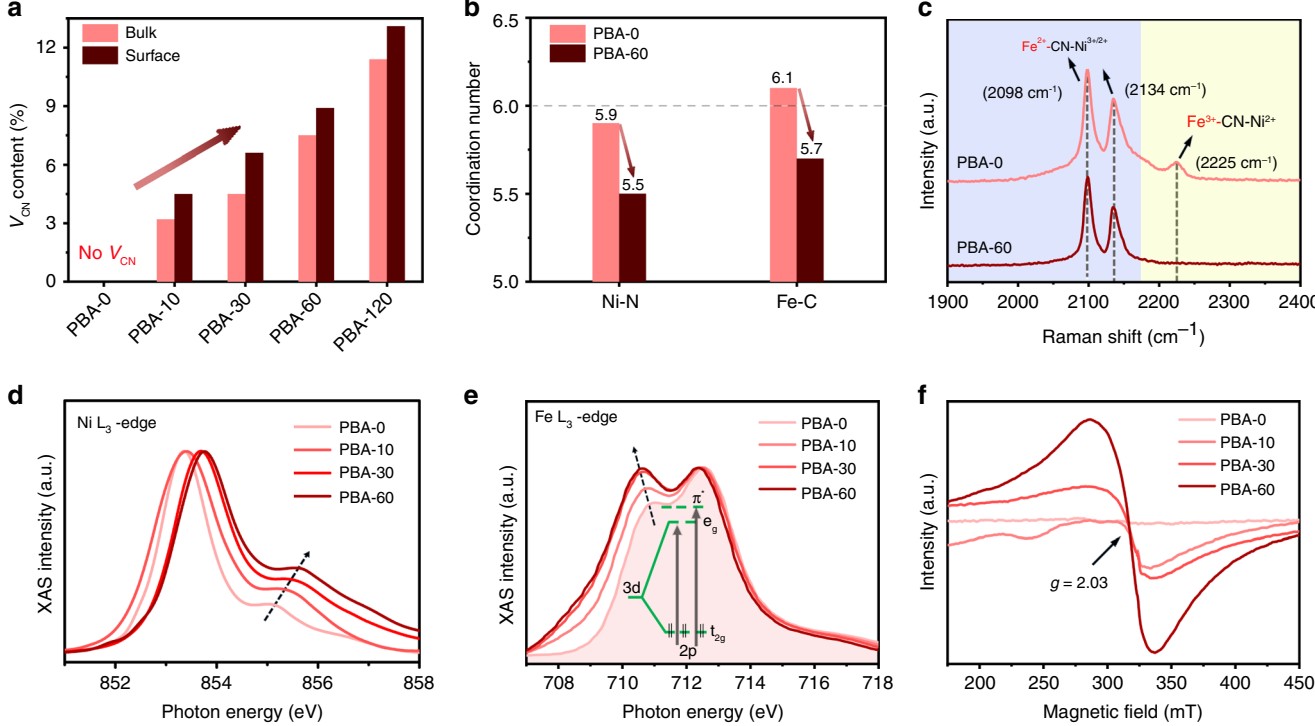

**Fig. 4** Chemical and structure alterations of the $V_{CN}$-mediated PBA catalysts. **a** The bulk and surface $V_{CN}$ contents of different $V_{CN}$-mediated PBA catalysts. **b** Comparison of Ni–N and Fe–C coordination numbers for PBA-0 and PBA-60 catalysts. **c** Raman spectra of PBA-0 and PBA-60 catalysts. **d**, **e** Ni $L_3$-edge and Fe $L_3$-edge XAS spectra of PBA-0 and different $V_{CN}$-mediated PBA catalysts, respectively. **f** ESR spectra of PBA-0 and different $V_{CN}$-mediated PBA catalysts

spectroscopy (EDX) analyses reveal that Fe suffers a certain loss at the first 5 h of cycling for PBA-60, which is retained thereafter even up to 25 h (Fig. 5b; see Supplementary Fig. 32 for the detailed Fe/Ni atomic ratios). In contrast, the PBA-0 loses most Fe in the structure after 25 h of continuous operation. These results agree well with our STEM elemental mapping studies shown in Fig. 5d–g. Moreover, the time-dependent Fourier transform infrared (FT-IR) spectra on PBA-60 catalyst exhibit that three bands at 3650 cm$^{-1}$, 880 cm$^{-1}$, and 640 cm$^{-1}$ become dominant with time (Fig. 5c; Supplementary Fig. 33), which belongs to the O–H stretching mode in brucite-like Ni(OH)$_2$ structure[45], as well as the Fe–O–H[46] and Ni–O–H[45] bending modes, respectively, indicating the formation of Ni–Fe oxy (hydroxide) active surface layer during OER. Note that the Fe–O–H bond at 880 cm$^{-1}$ was not detected for PBA-0 cycled for 25 h because of the loss of Fe, consistent with above results. The surface self-reconstruction process on our Ni–Fe PBA OER catalyst can be further confirmed by the Raman spectroscopy measurements (Supplementary Fig. 34) and the O K-edge XAS data (Supplementary Fig. 35).

On the basis of above investigations, we become clear about the origin of the outstanding OER properties on $V_{CN}$-mediated PBA catalysts. In the Ni–Fe PBA structure, Fe atoms are bonded to six C atoms. The removal of CN groups in PBAs makes the Fe–C coordination numbers decreased, leaving a number of coordinatively unsaturated Fe sites (Fig. 5h). These open metal centers are favorable to bond with oxygen to yield Fe–O bonds during OER, which therefore hampers the Fe loss into the electrolyte. Nevertheless, the fresh PBA-0 catalysts are subjected to a complete anion exchange between [Fe(CN)$_6$]$^{4-}$ and OH$^{-}$ in alkaline solution, which causes significant Fe leaching (Fig. 5h)[14,15]. Therefore, $V_{CN}$ can largely prevent the Fe loss from PBA catalysts, which leads to the formation of OER-active NiFeOOH surface layer and enhances the OER activity.

## Discussion

In conclusion, we have demonstrated that unconventional CN vacancies were generated in Ni–Fe PBA materials via ionized N$_2$ bombardment, as evidenced by multiple characterization techniques. Such $V_{CN}$ is largely distinct from the well-studied vacancies previously documented, which not only tunes the local electronic structure and coordination environment of the Ni–Fe sites, but also limits the loss of Fe element during OER process; these together enable a new, robust, and high-performance Ni–Fe oxyhydroxide from PBA pre-catalyst during OER. We anticipate that this unusual $V_{CN}$ could also be achievable in other PBAs, which thus opens up the possibilities for exploring new applications of PBAs beyond catalysis.

## Methods

**Material synthesis**. The NiMoO$_4$ nanorods were prepared by a hydrothermal method in a pure water system according to our previous works[29,30]. Briefly, 2 mmol Ni(NO$_3$)$_2$•6H$_2$O, 2 mmol Na$_2$MoO$_4$•2H$_2$O, and 35 mL of H$_2$O were mixed to form a clear solution. The mixture was transferred into a Teflon-lined stainless autoclave (50 mL) and heated at 150 °C for 6 h. After reaction, NiMoO$_4$ powder was obtained by centrifugation, which was then washed and dried for use. Next, 40 mg of NiMoO$_4$ powder was dispersed in the 20 mL K$_4$Fe(CN)$_6$•3H$_2$O (0.3 mmol) aqueous solution with ultrasonication. After 2 h of reaction with drastic stirring at room temperature, the product was centrifuged, washed, and dried to obtain PBA-0. The PBA-0 powder was treated by N$_2$ plasma at various irradiation times of 10 min, 30 min, 60 min, and 120 min with the plasma power of 100 W and the pressure of 3 mTorr. The synthesis of Co–Fe PBAs are the same with the synthesis of Ni–Fe PBAs, except for replacing the Ni(NO$_3$)$_2$•6H$_2$O with Co(NO$_3$)$_2$•6H$_2$O in the first step. The Co–Fe PBA samples were bombarded by N$_2$ plasma for different times, giving desired samples for comparison.

**Material characterizations**. Morphology of the samples was investigated by TEM (Hitachi H7650) and HRTEM (JEM-ARM 200 F). XRD pattern was obtained from a Philips X'Pert Pro Super X-ray diffractometer equipped using Cu Kα radiation (λ, 1.54184 Å). Raman spectra were carried out at a Raman microscope with a 514-nm excitation laser. Fourier transform infrared (FT-IR) spectra were recorded on a Fourier transform infrared spectrometer (Bruker Vector 22) with KBr disk method.

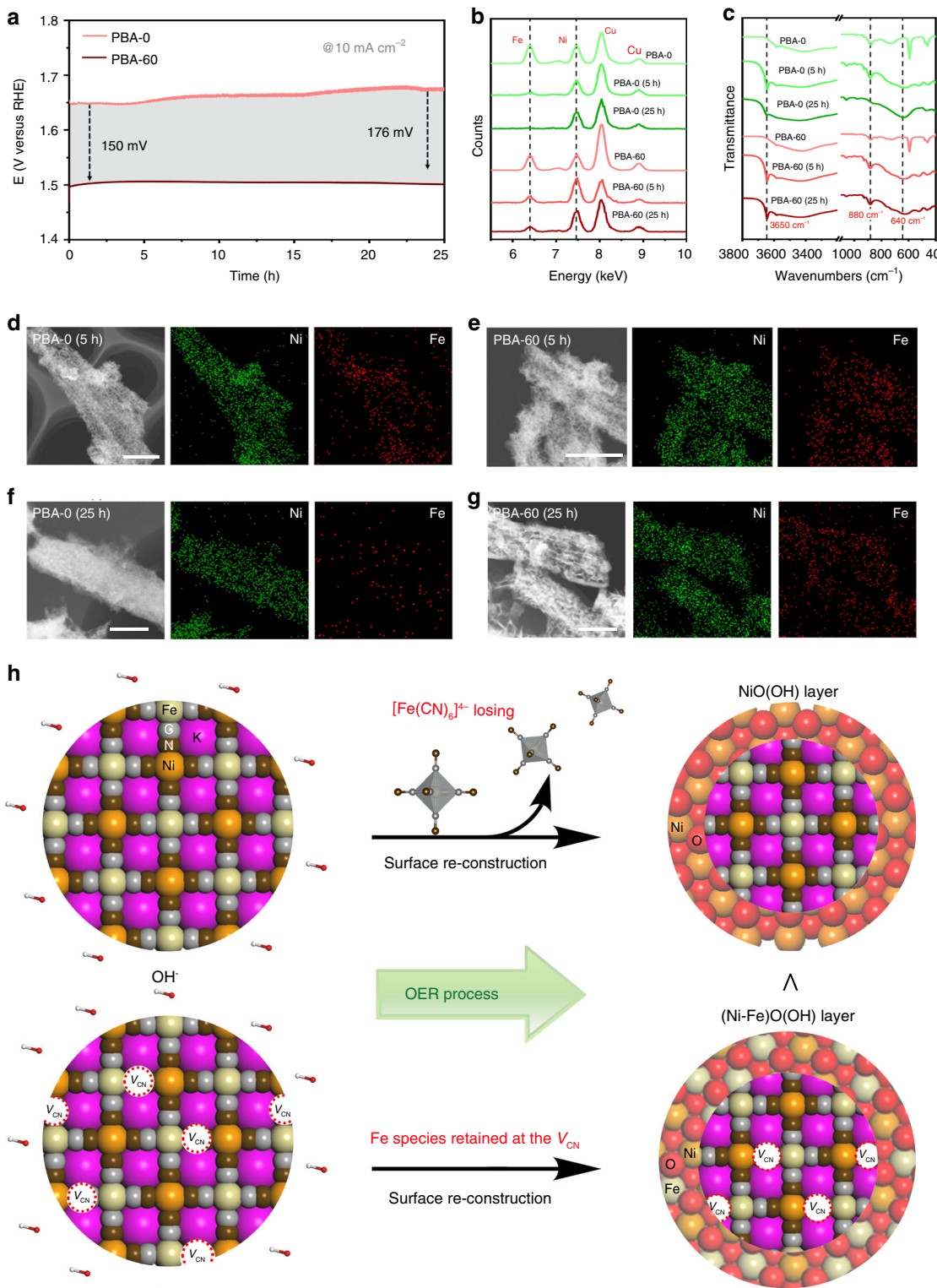

**Fig. 5** Performance stability and Fe-leaching suppression. **a** Chronopotentiometric responses recorded on PBA-0 and PBA-60 at a constant current density of 10 mA cm$^{-2}$. **b** EDX spectra of PBA-0 and PBA-60 catalysts that cycled for different times. The Cu signals come from Cu-based TEM grid. **c** FT-IR spectra of PBA-0 and PBA-60 catalysts that cycled for different times. **d, e** EDX elemental mappings of PBA-0 and PBA-60 catalysts that cycled for 5 h, respectively. Scale bars, 250 nm. **f, g** EDX elemental mappings of PBA-0 and PBA-60 catalysts that cycled for 25 h, respectively. Scale bars, 100 nm. **h** Illustrations of the surface reconstruction on the Ni–Fe PBA catalyst during OER, where the V$_{CN}$ enables the suppression of Fe loss, permitting the formation NiFeOOH surface active layer

Photoluminescence (PL) spectra were obtained from a Hitachi F-4600 fluorescence spectrophotometer with an excitation light of 300 nm. Cathodoluminescence (CL) images were obtained from SEM (Sirion200) with equipped with CL system under 10 keV. XPS was taken on an X-ray photoelectron spectrometer (ESCALab MKII) with an X-ray source (Mg Kα $h\nu = 1253.6$ eV). ICP data were obtained by an Optima 7300 DV instrument. $N_2$ adsorption/desorption analysis was conducted on an ASAP 2020 (Micromeritics, USA) at 77 K. ESR spectra were obtained from a JEOL JES-FA200 EPR spectrometer (9065.8 MHz, X band, 300 K). PAS were conducted with a fast-slow coincidence ORTEC system with a time resolution of about 230 ps full width at half-maximum, and our samples were pressed into about 1 -mm-thick disk with a 5 mCi source of $^{22}$Na sandwiched between two identical sample disks. The X-ray absorption spectra of Ni and Fe K-edges were obtained from the beamline 1W1B station of Beijing Synchrotron Radiation Facility (China). The X-ray absorption spectra of Ni L-edges and O K-edges were obtained from the 4B9B beamline of Beijing Synchrotron Radiation Facility (China). The X-ray absorption spectra of Fe L-edges were performed on the BL10B beamline of National Synchrotron Radiation Laboratory in Hefei (China).

**Electrochemical measurements**. The standard three-electrode system was applied to perform the OER electrochemical measurements on the IM6ex electrochemical workstation (ZAHNER elektrik, Germany). A rotating disk electrode (RDE) made of glassy carbon (PINE, 5 -mm diameter, disk area: 0.196 cm$^2$) was used as the working electrode. Platinum foil was used as the counter electrode, and the Ag/AgCl (3.5 M KCl) was used as the reference electrode. The potential difference between Ag/AgCl and RHE was calibrated by the cyclic voltammetry test in H$_2$-saturated 1 M KOH electrolyte ($E_{RHE} = E_{Ag/AgCl} + 1.03$ V).

To make the working electrodes, 5 mg of catalyst powder was dispersed in 960 μL of ethanol with 40 μL of Nafion solution (5 wt%), which was then ultrasonicated to yield catalyst ink. A uniform catalyst film was obtained by pipetting 10 μL of catalyst ink onto the GC electrode, leading to the catalyst loading of ~0.255 mg cm$^{-2}$. Before OER measurements, the electrolyte (1 M KOH) was bubbled with O$_2$ gas for at least 30 min. The electrodes were pre-cycled between 0 and 0.8 V vs. Ag/AgCl at a sweep rate of 100 mV s$^{-1}$ for 20 cycles until reaching the stable state, then the OER polarization curves were recorded at a sweep rate of 5 mV s$^{-1}$ (Supplementary Fig. 11). The EIS measurement was performed over a frequency ranging from 100 KHz to 100 mHz at the amplitude voltage of 5 mV. For the long-term stability test, the catalyst was supported on the carbon fiber paper (1 mg cm$^{-2}$), which was directly used as the working electrode. All the polarization curves were corrected with iR-correction unless otherwise stated. The onset potential is defined as the potential that yields a current density of 1 mA cm$^{-2}$.

## Data availability
The data that support the findings of this study are available from the corresponding authors upon request.

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

## Acknowledgements

This work was supported by the funding support from the National Natural Science Foundation of China (Grants 21521001, 21431006, 21225315, 21321002, 91645202, and 51702312), the Users with Excellence and Scientific Research Grant of Hefei Science Center of CAS (2015HSCUE007), the Key Research Program of Frontier Sciences, CAS (Grant QYZDJ-SSW-SLH036), the Chinese Academy of Sciences (Grants KGZD-EW-T05, XDA090301001), the Fundamental Research Funds for the Central Universities (WK2060190045, WK2340000076), and the Recruitment Program of Global Youth Experts. Z.-Y.Y. acknowledges the China Postdoctoral Science Foundation (2018M632545). This work was partially carried out at the USTC Center for Micro and Nanoscale Research and Fabrication.

## Author contributions

M.-R.G., Z.-Y.Y. and S.-H.Y. conceived the idea. S.-H.Y. and M.-R.G. supervised the project. Z.-Y.Y., Y.D. and Y.C. performed the experiments, collected and analyzed the data. T.M. and Y.L. provided the help to analyze the results. J.-D.L. and B.-J.Y. collected and analyzed the PAS data. X.-K.L., W.L. and T.Y. collected and analyzed the XANES data. X.-S.Z. and J.-F.Z. collected and analyzed the XAS data. Z.-Y.Y., M.-R.G. and S.-H.Y. co-wrote the paper. All authors discussed the results and commented on the paper.

## Additional information

**Competing interests:** The authors declare no competing interests.

