## [Peer Review File · Nature Communications]

Reviewers' Comments:

Reviewer #1:

Remarks to the Author:

The authors reported an approach to yield CN defects in PBAs through ionized nitrogen plasma treatment and subsequently investigated the role of these defects played in promoting the electrocatalytically performance. They found that CN vacancies are able to suppress the leaking of Fe ions during the electrochemical process, and thus in-situ generates highly-active (Ni,Fe)OOH species on the surface towards OER. The result is novel and important for designing and understanding OER electrocatalysts. However, several characterizations are misunderstanding and the evidences are not enough. I recommend the paper to be published in your journal after a major revision.

1. Electrochemical characterizations

1) It is found that the pristine PBA exhibits very poor activity, i.e., 440 mV at 10 mA cm⁻². This value is not consistent with the result in previous work, as the cited reference 15. The authors should describe the details of electrochemical characterizations.

2) Generally, the OER electrocatalysts have an activation time because of the "self-reconstruction". The LSV curves should be measured after the electrocatalyst become stable. How long is the activation time?

3) The Ni-based OER electrocatalysts, especially for (Ni,Fe)OOH, an intense redox peak should be appear (J. Am. Chem. Soc. 2015, 137, 15090–15093; J. Am. Chem. Soc. 2016, 138, 5603–5614; J. Am. Chem. Soc. 2017, 139, 2070–2082). The authors should explain the absence of this significant feature of LSV curves since that they demonstrated that (Ni,Fe)OOH is active species.

2. In Figure 4d and 4e, the author used soft X-ray spectroscopy to identify the oxidation of transition-metals. In particular, they found Ni²⁺ ions in pristine PBA are oxidized to be Ni³⁺ ions in the presence of CN vacancies through the shift of Ni L-edge XAS. Actually, this judgment is misunderstanding. The L-edge XAS of TM ions with different oxidation state should have distinct shape, which arises from multiplet effect, as showed in the cited reference 36. Ni³⁺ ion L-edge XAS usual contains a double-peak shape (PHYSICAL REVIEW B 76, 195122 2007). In terms of presented L-edge XAS, it is concluded that Ni ions remained to be +2 oxidation state.

3. The authors identified in-situ generated (Ni,Fe)OOH as active species. On the other hand, the author mentioned that "the oxidation state of Ni in PBAs was boosted to 3+ through VCN mediation (Fig. 4d and Fig. 4f), which can be devoted to advance the OER performance because it is more readily oxidized to the most active Ni⁴⁺ species." Here, this oxidation of Ni ions occurs in PBAs, rather than (oxy)-hydroxides. These descriptions are confusing. Actually, the deprotonation reaction during the OER have been unraveled, leading to that Ni²⁺ can be readily transform into active Ni⁴⁺ ions in (Ni,Fe)OOH (J. Am. Chem. Soc. 2015, 137, 15112–15121). According to the presented work, I suppose that the primary role of CN vacancies played is suppressing the leaking of Fe ions, rather than generating Ni³⁺ ions.

4. The author demonstrated a (Ni,Fe)OOH active layer after OER. To make it more convincible, O K-edge XAS should be provided.

Reviewer #2:

Remarks to the Author:

This manuscript reports the synthesis and the remarkable electrocatalytic activity of a Ni-Fe oxhydroxide phase towards OER in alkaline media. The work is quite exhaustive, and (mostly)

technically correct. The results are indeed of merit, and they certainly deserve to be published in Nature Communications. However, the manuscript needs some major corrections since the text includes some misconceptions, wrong assumptions, and inaccurate comparison.

If the authors are able to revise the text to present their results in their right value, I will be happy to support publication.

My major concerns are:

1) In the title, and through most of the text, the authors give the wrong impression about the actual nature of their catalyst. Most readers will think that the catalyst is a Fe-Ni Prussian blue derivative. It is not until the very end, in Figure 5, when the authors properly describe the operando transformation of the Prussian blue into a Fe-Ni oxohydroxide, the true and genuine catalysts. This is not a surprise since Fe-Ni PB is not stable in highly alkaline media, as the authors implicitly explain .

So, the authors should make very clear from the beginning of the text, including the abstract and title, that the PB material is a pre-catalysts, and that a Fe-Ni oxohydroxide is indeed the genuine catalyst. This is not a demerit to the work, since many publications deal with novel processing techniques to improve the activity of Fe-Ni oxohydroxides. But in the present case, the authors are confusing the reader. This must be solved before publication.

For instance, I would propose a title in this line: "Unconventional CN vacancies formed in a Prussian blue pre-catalyst boost oxygen evolution reaction on Fe-Ni oxyhydroxides"

With further modifications in the abstract and introduction.

2) In the same line, the authors make a wrong assumption using IrOX as state-of-the-art OER catalyst for comparison. This is not only the authors fault, since many manuscripts make the same mistake. But this should be stopped. The state of the art OER catalyst in alkaline conditions is certainly the family of Fe-Ni oxohydroxides. So the authors should compare their results with those of the same family. If they need recent references, I would suggest Acc Them Res 2018, 51, 1571; Energy Environ Sci 2018, 11, 2858; Small 2017, 13, 1701931; Nano Res 2015, 8, 23. The results should be in the right context.

3) The third wrong assumption is the identification of the Ni centres as catalytically active. On page 12, they authors write: "Although debate still exists, prior studies have described that, for Ni-Fe oxides, the in-situ formed Ni⁴⁺ species during OER process are the most active sites^{38,39}. The presence of Fe in the structure was thought to be able to increase the conductivity of NiOOH and also affect its electronic structure by partial-charge transfer^{38,39}." Well, the most recent literature certainly points towards Fe being the active site. The authors may use Molecules 2018, 23, 903 as an excellent reference to support this fact.,

In summary, I think the authors have found a nice (although complex) processing route to obtain highly active Ni-Fe oxohydroxyde porous particles by in situ evolution of a PB material. It is particularly relevant that the CN vacancies allow for the formation of the Ni-Fe oxohydroxyde, avoiding the natural disolución of the PB in alkaline conditions, which yields a very different iron-defective oxohydroxyde. But for this manuscript to be accepted, the authors need to put the results in the right context, and to describe their findings in a precise and accurate way, to avoid misinterpretations.

We thank all the reviewers for their valuable comments and questions that help us significantly improve the revised manuscript.

REVIEWER REPORTS:

Reviewer #1 (Remarks to the Author):

The authors reported an approach to yield CN defects in PBAs through ionized nitrogen plasma treatment and subsequently investigated the role of these defects played in promoting the electrocatalytically performance. They found that CN vacancies are able to suppress the leaking of Fe ions during the electrochemical process, and thus in-situ generates highly-active (Ni,Fe)OOH species on the surface towards OER. The result is novel and important for designing and understanding OER electrocatalysts. However, several characterizations are misunderstanding and the evidences are not enough. I recommend the paper to be published in your journal after a major revision.

Response: We greatly appreciate the reviewer for the positive feedbacks on the contents presented in our manuscript.

1. Electrochemical characterizations

1) It is found that the pristine PBA exhibits very poor activity, i.e., 440 mV at 10 mA cm⁻². This value is not consistent with the result in previous work, as the cited reference 15. The authors should describe the details of electrochemical characterizations.

*Response: Thanks for the reviewer's careful comments. In the reference 15 (J. Am. Chem. Soc. 2018, 140, 11286), the authors mixed the NiFe-PBA catalyst with carbon black (mass ratio of 1:1) and measured the OER activity of the resultant NiFe-PBA/carbon black. They observed a 410 mV overpotential at 10 mA cm⁻² for first cycle, which was decreased to 258 mV after 100 CV cycles. For our OER evaluation, we tested the OER activity of the pristine NiFe-PBA directly without adding any conductive additives. Therefore, the value reported in our manuscript reflects the intrinsic OER activity of NiFe-PBA, without any contribution from conductive additives as demonstrated in the reference 15. We further note that our measured value (440 mV at 10 mA cm⁻²) for the pristine NiFe-PBA agrees well with values of other NiFe-PBA-based catalysts documented previously, such as ~420 mV for NiFe-PBA cubes (ACS Appl. Mater. Interfaces, 2017, 9, 26134), 430 mV for NiFe-PBA cubes (Adv. Mater., 2017, 29, 1703870), and ~470 mV for NiCoFe-PBA cubes (Electrochim. Acta, 2019, 299, 575). Following your suggestion, we have provided details on electrochemical characterizations in the revised manuscript (see the **revised Main Text, Pages 17-18**).*

2) Generally, the OER electrocatalysts have an activation time because of the “self-reconstruction”. The LSV curves should be measured after the electrocatalyst become stable. How long is the activation time?

*Response: Thanks for the careful comments and question. As to our NiFe-PBA catalyst, we also observed a “self-reconstruction” process that activates the OER performance. We cycled the catalyst between 0 and 0.8 V vs. Ag/AgCl at a sweep rate of 100 mV s⁻¹ for specified cycles and then recorded the LSV curves at a sweep rate of 5 mV s⁻¹ (see **Figure R1**). We saw that both PBA-0 and PBA-60 become stable after around 20 CV cycles, further cycling the catalysts will not alter the polarization curves. Hence the activation process for our catalysts is pretty fast. We have added the new data in the revised manuscript and provided some discussions accordingly (see the **revised Main Text, Pages 7 and 18, and Supplementary Fig. 11**).*

Figure R1. OER polarization curves of PBA-0 and PBA-60 recorded at a sweep rate of 5 mV s⁻¹. The polarization curves were recorded after given number of cycles between 0 and 0.8 V vs. Ag/AgCl at a sweep rate of 100 mV s⁻¹.

3) The Ni-based OER electrocatalysts, especially for (Ni,Fe)OOH, an intense redox peak should be appear (J. Am. Chem. Soc. 2015, 137, 15090–15093; J. Am. Chem. Soc. 2016, 138, 5603–5614; J. Am. Chem. Soc. 2017, 139, 2070–2082). The authors should explain the absence of this significant feature of LSV curves since that they demonstrated that (Ni,Fe)OOH is active species.

*Response: We thank the reviewer for the insightful comments and suggestion. Consistent with previous reports, we for sure observed the redox peak for our each NiFe-PBA catalyst. In our original manuscript, we showed the polarization curves between 1.43 and 1.67 V vs. RHE, aiming to give a better comparison of the onset potential of the studied catalyst. However, the redox peaks (Ni^{2+/3+}) of these catalysts appear before 1.4 V vs. RHE (see **Figure R2** below), which cause the “absence” of redox peaks in our original manuscript. As the reviewer comments, such redox peaks always appear for Ni-based and NiFe-based OER catalysts (J. Am. Chem. Soc. 2015, 137, 15090; J. Am. Chem. Soc. 2016, 138, 5603; J. Am. Chem. Soc. 2017, 139, 2070). Our results perfectly agree with previous data. In the revised manuscript, we have modified the related figures and provided*

some discussions there (see the revised Main Text, Page 7, Fig. 3a).

Figure R2. OER polarization curves of different studied Ni-Fe PBA catalysts.

2. In Figure 4d and 4e, the author used soft X-ray spectroscopy to identify the oxidation of transition-metals. In particular, they found Ni^{2+} ions in pristine PBA are oxidized to be Ni^{3+} ions in the presence of CN vacancies through the shift of Ni L-edge XAS. Actually, this judgment is misunderstanding. The L-edge XAS of TM ions with different oxidation state should have distinct shape, which arises from multiplet effect, as showed in the cited reference 36. Ni^{3+} ion L-edge XAS usual contains a double-peak shape (PHYSICAL REVIEW B 76, 195122 2007). In terms of presented L-edge XAS, it is concluded that Ni ions remained to be +2 oxidation state.

Response: We thank the reviewer for the insightful comments. We previously conducted the Ni L-edge XAS on the BL10B beamline of National Synchrotron Radiation Laboratory in Hefei (China). After discussing with the beamline scientist, we realized that the energy and resolution of Hefei Light Source BL10B beamline is relatively low, which resulted in the single-peak shape of Ni L_3 -edge in the original manuscript. We note that similar observations on this beamline have been documented in previous reports (for examples: Angew. Chem. Int. Ed. 2018, 57, 15445; Adv. Mater. 2017, 29, 1701687).

*To gather high-resolution Ni L-edge XAS spectra, we contacted with the Beijing Light Source 4B9B beamline and measured samples over there (see **Figure R3**). We purchased high-purity NiO from Sigma-Aldrich as the Ni^{2+} reference. We chose the LiNiO_2 as the Ni^{3+} reference because it contains Ni^{3+} (Nat. Chem. 2018, 10, 149; Phys. Rev. B 2007, 76, 195122). The LiNiO_2 reference was synthesized by a solid state method according to Tong's work (Chem. Commun. 2016, 52, 4239). It is known that Ni L-edge spectra correspond to the Ni $2p \rightarrow \text{Ni } 3d$ transitions and are split by the Ni $2p$ spin-orbit interaction, resulting in the Ni $2p_{3/2}$ (L_3 -edge) and Ni $2p_{1/2}$ (L_2 -edge) regions. Further splitting in these regions occurs due to the Ni $2p-3d$ electrostatic interactions and crystal field effects. Our new Ni L_3 -edge XAS spectra clearly uncover the double-peak shape (see **Figure R3**).*

Figure R3b gives the magnified Ni L_3 -edges, from which we observed the strongest shoulder peak for LiNiO_2 reference (at 855~856 eV), indicating rich Ni^{3+} component in the structure (Nat. Chem. 2018, 10, 149; Phys. Rev. B 2007, 76, 195122). From **Figure R3b**, we also observed the gradually increased intensity of this shoulder peak from PBA-0 (also NiO reference) to PBA-60, suggesting that surface Ni^{2+} ions were oxidized to Ni^{3+} . These new results thus demonstrate that Ni in our NiFe-PBA catalyst indeed partially oxidized to Ni^{3+} after N_2 plasma bombardment. We have added these new data in the revised manuscript and provided some discussions accordingly (see the revised Main Text, Pages 11,17, Fig. 4d, and Supplementary Fig. 25).

Figure R3. Ni L-edge XAS spectra. **a**, The Ni L-edge spectra of different studied samples with NiO and LiNiO_2 as references. **b**, The magnified Ni L_3 -edge spectra.

3. The authors identified in-situ generated (Ni,Fe)OOH as active species. On the other hand, the author mentioned that “the oxidation state of Ni in PBAs was boosted to 3+ through V_{CN} mediation (Fig. 4d and Fig. 4f), which can be devoted to advance the OER performance because it is more readily oxidized to the most active Ni^{4+} species.” Here, this oxidation of Ni ions occurs in PBAs, rather than (oxy)-hydroxides. These descriptions are confusing. Actually, the deprotonation reaction during the OER have been unraveled, leading to that Ni^{2+} can be readily transform into active Ni^{4+} ions in (Ni,Fe)OOH (J. Am. Chem. Soc. 2015, 137, 15112–15121). According to the presented work, I suppose that the primary role of CN vacancies played is suppressing the leaking of Fe ions, rather than generating Ni^{3+} ions.

*Response: We thank the reviewer for the thoughtful comments. In the original manuscript, we observed that the oxidation state of Ni in PBA catalysts was partially increased to 3+ after the formation of V_{CN} in the structure (also see our new high-resolution Ni L-edge XAS spectra in **Figure R3**). We thus propose that such V_{CN} -mediated PBA catalyst may be easier to generate Ni^{4+} species during the OER process, because oxidizing Ni^{3+} to Ni^{4+} is more readily to occur. In the revised manuscript, we want to quit this assumption because Ni^{2+} also can be readily transform into Ni^{4+} species, as described in J. Am. Chem. Soc. 2015, 137, 15112 that suggested by the reviewer. Accordingly, we have modified the related discussions very carefully to avoid any misunderstanding*

(see the revised Main Text, Pages 12-13).

4. The author demonstrated a (Ni,Fe)OOH active layer after OER. To make it more convincible, O K-edge XAS should be provided.

Response: We thank the reviewer for this useful suggestion. Following your suggestion, we collected the O K-edge XAS spectra of various studied samples (see Figure R4). We also synthesized NiFe-based layered double hydroxide (NiFe-LDH) as reference according to Dai's work (J. Am. Chem. Soc. 2013, 135, 8452; see the revised Supplementary Figure 12 for the characterization). The NiFe-oxide was gained by annealing NiFe-LDH at 350 °C in air. The NiFe-OOH reference was achieved by electrochemically treating NiFe-LDH at 1.8 V vs. RHE for 10 h under OER condition (J. Am. Chem. Soc. 2013, 135, 12329). Compared to NiFe-LDH and NiFe-oxide, NiFe-OOH reference exhibited obvious features arising from -O and -OH species in O K-edge XAS spectra (see Figure R4). As demonstrated in Figure R4, our new O K-edge XAS spectra clearly show the signals from -O and -OH species for the PBA-0 and PBA-60 after OER tests, which evidence that the oxy(hydroxide) active layer was formed on the surface of PBA-0 and PBA-60 during OER process. We have added this data and related discussions in the revised manuscript (see the revised Main Text, Page 14, and Supplementary Fig. 35).

Figure R4. O K-edge XAS spectra of various studied samples with NiFe-LDH, NiFe-oxide and NiFe-OOH as references.

Reviewer #2 (Remarks to the Author):

This manuscript reports the synthesis and the remarkable electrocatalytic activity of a Ni-Fe oxohydroxide phase towards OER in alkaline media. The work is quite exhaustive, and (mostly) technically correct. The results are indeed of merit, and they certainly deserve to be published in Nature Communications. However, the manuscript needs some major corrections since the text includes some misconceptions, wrong assumptions, and inaccurate comparison. If the authors are

able to revise the text to present their results in their right value, I will be happy to support publication.

Response: We greatly appreciate the reviewer's high praise and support on the publication of this work.

1) In the title, and through most of the text, the authors give the wrong impression about the actual nature of their catalyst. Most readers will think that the catalyst is a Fe-Ni Prussian blue derivative. It is not until the very end, in Figure 5, when the authors properly describe the operando transformation of the Prussian blue into a Fe-Ni oxohydroxide, the true and genuine catalysts. This is not a surprise since Fe-Ni PB is not stable in highly alkaline media, as the authors implicitly explain. So, the authors should make very clear from the beginning of the text, including the abstract and title, that the PB material is a pre-catalysts, and that a Fe-Ni oxohydroxide is indeed the genuine catalyst. This is not a demerit to the work, since many publications deal with novel processing techniques to improve the activity of Fe-Ni oxohydroxides. But in the present case, the authors are confusing the reader. This must be solved before publication.

For instance, I would propose a title in this line: "Unconventional CN vacancies formed in a Prussian blue pre-catalyst boost oxygen evolution reaction on Fe-Ni oxyhydroxides". With further modifications in the abstract and introduction.

Response: We greatly appreciate the reviewer for sharing his/her insightful thoughts regarding the expression about the real catalyst reported in this work. We accept the review's critical comments. Following your comments and suggestion, we have carefully modified the content in the manuscript, including the Title, Abstract, Introduction and Conclusions.

We agree with our two reviewers that the key finding in our manuscript is that the nonvel CN vacancies can suppress the Fe-leaching in PBA pre-catalyst during OER. We thus think that this key merit should be reflected in the Title. Also, based on the Title suggestion from the reviewer, we have changed the Title as "Unconventional CN vacancies suppress iron-leaching in Prussian blue analogue pre-catalyst for boosted oxygen evolution catalysis". In this Title, we highlighted that the Prussian blue is actually "pre-catalyst", which will eliminate the misunderstanding.

We believe that our revised manuscript can clearly distinguish the NiFe-PBA pre-catalyst and the actual Ni-Fe oxyhydroxide catalyst, thus avoiding any uncertainty and misunderstanding to the audiences. (see the revised Main Text, Pages 1,2,4,16).

2) In the same line, the authors make a wrong assumption using IrO_x as state-of-the-art OER catalyst for comparison. This is not only the authors fault, since many manuscripts make the same mistake. But this should be stopped. The state of the art OER catalyst in alkaline conditions is certainly the

family of Fe-Ni oxyhydroxides. So the authors should compare their results with those of the same family. If they need recent references, I would suggest Acc Them Res 2018, 51, 1571; Energy Environ Sci 2018, 11, 2858; Small 2017, 13, 1701931; Nano Res 2015, 8, 23. The results should be in the right context.

*Response: We thank the reviewer for the useful suggestion. We agree with the reviewer that some Fe-Ni oxyhydroxides actually surpass the activity IrO_x catalyst in alkaline electrolyte. Considering that many Fe-Ni oxyhydroxide OER catalysts have been developed, we carefully selected the most credible Fe-Ni oxyhydroxide reported previously as the reference. We noticed that the Fe-Ni oxyhydroxide developed by Dai's group is very reliable and active (J. Am. Chem. Soc. 2013, 135, 8452; **Times cited: 1304**). We thus prepared such NiFe-based layered double hydroxide (NiFe-LDH) by following Dai's method. Our TEM image and XRD pattern of the as-synthesized NiFe-LDH show the nanoplate morphology with α -Ni(OH)₂ phase (see **Figure R5**), perfectly matching with Dai's results.*

Figure R5. Characterization of NiFe-LDH. **a**, TEM image of NiFe-LDH. **b**, XRD pattern of NiFe-LDH.

Following your suggestion, we compared the OER activity of the representative NiFe-LDH reference with our NiFe-PBA catalysts (see **Figure R6**). We found that the optimal PBA-60 catalyst requires 283 mV overpotential at 10 mA cm⁻², whereas 315 mV overpotential is required for NiFe-LDH reference. We note that the 315 mV overpotential at 10 mA cm⁻² for NiFe-LDH reference is comparable to most of NiFe-based OER catalysts reported previously, such as 347 mV for bulk NiFe-LDH and 302 mV for exfoliated NiFe-LDH (Nat. Commun. 2014, 5, 4477); 310 mV for NiFe-OOH nanoparticles (J. Am. Chem. Soc. 2017, 139, 2070); 325 mV for NiFe-LDH nanosheets (Adv. Mater. 2018, 30, 1705106); 347 mV for NiFe-LDH nanoflakes (Nat. Commun. 2016, 7, 11981) and 350 mV for NiFeO_x nanofilms (J. Am. Chem. Soc. 2013, 135, 16977). We thus highlight that our PBA-60 catalyst not only surpasses the NiFe-LDH reference, but also exceeds most of NiFe-based OER catalysts reported previously. Moreover, at overpotentials of 300, 320 and 340 mV, PBA-60 has 2.7, 4.5 and 8-fold improvements in current density versus NiFe-LDH, respectively (**Figure R6c**). Besides, our PBA-60 catalyst also performs superior to the NiFe-LDH reference in terms of Tafel

slope, charge transfer resistance and activation energy (Figure R6b, d and e). All the above results together clearly illustrate that our PBA-60 catalyst is a better-performing OER catalyst. We have added these new data in the revised manuscript and provided some discussions (see the revised Main Text, Pages 7-9, Fig. 3 and Supplementary Fig. 12).

Figure R6. Comparison of the OER performance between our developed PBA catalysts and the NiFe-LDH reference.

3) The third wrong assumption is the identification of the Ni centres as catalytically active. On page 12, they authors write: “Although debate still exists, prior studies have described that, for Ni-Fe oxides, the in-situ formed Ni⁴⁺ species during OER process are the most active sites^{38,39}. The presence of Fe in the structure was thought to be able to increase the conductivity of NiOOH and also affect its electronic structure by partial-charge transfer^{38,39}.” Well, the most recent literature certainly points towards Fe being the active site. The authors may use Molecules 2018, 23, 903 as an excellent reference to support this fact.

Response: We thank the reviewer for the useful comments and suggestion. On the basis of the comments, we revisited very recent literatures such as J. Am. Chem. Soc. 2018, 140, 7748 and Molecules 2018, 23, 903 that recommended by the reviewer. We realized that, although the element nature of the active site in NiFe oxides is still highly debated, researchers prefer to believe that Fe center is the active site, whereas Ni site serves as an electrically conductive and chemically stable host for the Fe sites. Accordingly, we have modified the related discussions in the revised manuscript (see the revised Main Text, Page 12). We thank the reviewer again for pointing out this uncertainty.

In summary, I think the authors have found a nice (although complex) processing route to obtain highly active Ni-Fe oxohydroxyde porous particles by in situ evolution of a PB material. It is particularly relevant that the CN vacancies allow for the formation of the Ni-Fe oxohydroxyde, avoiding the natural disolución of the PB in alkaline conditions, which yields a very different iron-defective oxohydroxyde. But for this manuscript to be accepted, the authors need to put the results in the right context, and to describe their findings in a precise and accurate way, to avoid misinterpretations.

Response: Again, we greatly appreciate the reviewer's high praise and support on this work. On the basis of your insightful and helpful comments, we have carefully revised this manuscript to eliminate any misinterpretations and uncertainties, which now is suitable to the audience of Nature Communications.

Reviewers' Comments:

Reviewer #1:

Remarks to the Author:

The authors have fully addressed my questions and provided new experimental evidences to support their conclusion. In the regard of the novel finding, I recommend it to be published on Nature Communications.

Reviewer #2:

Remarks to the Author:

The authors have addressed all issues raised by both reviewers, and modify the text accordingly. The revised version is much improved, and all problems with partial data and wrong descriptions have been solved.

Therefore I am happy to support publication of the revised version. No further revisions are needed.

P. S. The point-to-point answers to the referees' comments

REVIEWERS' COMMENTS:

Reviewer #1 (Remarks to the Author):

The authors have fully addressed my questions and provided new experimental evidences to support their conclusion. In the regard of the novel finding, I recommend it to be published on Nature Communications.

Response: We thank the reviewer for strong support on the publication of this work.

Reviewer #2 (Remarks to the Author):

The authors have addressed all issues raised by both reviewers, and modify the text accordingly. The revised version is much improved, and all problems with partial data and wrong descriptions have been solved.

Therefore I am happy to support publication of the revised version. No further revisions are needed.

Response: We thank the reviewer for strong support on the publication of this work.